# Potts Relaxations and Soft Self-labeling for Weakly-Supervised Segmentation

## Abstract

We consider weakly supervised segmentation where only a fraction of pixels have ground truth labels (scribbles) and focus on a self-labeling approach where soft pseudo-labels on unlabeled pixels optimize some relaxation of the standard unsupervised CRF/Potts loss. While WSSS methods can directly optimize CRF losses via gradient descent, prior work suggests that higher-order optimization can lead to better network training by jointly estimating pseudo-labels, e.g. using discrete graph cut sub-problems. The inability of hard pseudo-labels to represent class uncertainty motivates the relaxed pseudo-labeling. We systematically evaluate standard and new CRF relaxations, neighborhood systems, and losses connecting network predictions with soft pseudo-labels. We also propose a general continuous sub-problem solver for such pseudo-labels. Soft self-labeling loss combining the log-quadratic Potts relaxation and collision cross-entropy achieves state-of-the-art and can outperform full pixel-precise supervision on PASCAL.

## 1 Introduction

Full supervision for semantic segmentation requires thousands of training images with complete pixel-accurate ground truth masks. Their high costs explain the interest in weakly-supervised approaches based on image-level class *tags* [21, 4], pixel-level *scribbles* [26, 36, 35], or *boxes* [23]. This paper is focused on weak supervision with *scribbles*, which we also call *seeds* or *partial masks*. While only slightly more expensive than image-level class tags, scribbles on less than $3\%$ of pixels were previously shown to achieve accuracy approaching full supervision without any modifications of the segmentation models. In contrast, tag supervision typically requires highly specialized systems and complex multi-stage training procedures, which are hard to reproduce. Our interest in the scribble-based approach is motivated by its practical simplicity and mathematical clarity. The corresponding methodologies are focused on the design of unsupervised or self-supervised loss functions and stronger optimization algorithms. The corresponding solutions are often general and can be used in different weakly-supervised applications.

### 1.1 Scribble-supervised segmentation

Assume that a set of image pixels is denoted by $\Omega$ and a subset of pixels with ground truth labels is $S \subset \Omega$, which we call *seeds* or *scribbles* as subset $S$ is typically marked by mouse-controlled UI for image annotations, e.g. see seeds over an image in Fig.7(a). The ground truth label at any given pixel $i \in S$ is an integer

$$\bar{y}_i \in \{1, \ldots, K\} \tag{1}$$

Submitted to 38th Conference on Neural Information Processing Systems (NeurIPS 2024). Do not distribute.

where $K$ is the number of classes including the background. Without much ambiguity, it is convenient to use the same notation $\bar{y}_i$ for the equivalent *one-hot* distribution

$$\bar{y}_i \;\equiv\; (\bar{y}_i^1, \ldots, \bar{y}_i^K) \in \Delta_{0,1}^K \qquad \text{for} \quad \bar{y}_i^k := [k = \bar{y}_i] \;\in \{0,1\} \tag{2}$$

where $[\,\cdot\,]$ is the *True* operator for the condition inside the brackets. Set $\Delta_{0,1}^K$ represents $K$ possible one-hot distributions, which are vertices of the $K$-class *probability simplex*

$$\Delta^K \;\; := \;\; \{p = (p^1, \ldots, p^K) \mid p^k \geq 0, \; \sum_{k=1}^{K} p^k = 1\}$$

representing all $K$-categorical distributions. The context of specific expressions should make it obvious if $\bar{y}_i$ is a class index (1) or the corresponding one-hot distribution (2).

Loss functions for weakly supervised segmentation with scribbles typically use *negative log-likelihoods* (NLL) over scribbles $i \in S \subset \Omega$ with ground truth labels $\bar{y}_i$

$$-\sum_{i \in S} \ln \sigma_i^{\bar{y}_i} \tag{3}$$

where $\sigma_i = (\sigma_i^1, \ldots, \sigma_i^K) \in \Delta^K$ is the model prediction at pixel $i$. This loss is a standard in full supervision where the only difference is that $S = \Omega$ and usually, no other losses are needed for training. However, in a weakly supervised setting the majority of pixels are unlabeled, and unsupervised losses are needed for $i \notin S$.

The most common unsupervised loss in image segmentation is the Potts model and its relaxations. It is a pairwise loss defined on pairs of *neighboring* pixels $\{i,j\} \in \mathcal{N}$ for a given neighborhood system $\mathcal{N} \subset \Omega \times \Omega$, typically corresponding to the *nearest-neighbor* grid (NN) [6, 17], or other *sparse* (SN) [38] and *dense* neighborhoods (DN) [22]. The original Potts model is defined for discrete segmentation variables, e.g. as in

$$\sum_{\{i,j\} \in \mathcal{N}} P(\sigma_i, \sigma_j) \qquad \text{where} \quad P(\sigma_i, \sigma_j) = [\sigma_i \neq \sigma_j]$$

assuming integer-valued one-hot predictions $\sigma_i \in \Delta_{0,1}^K$. This *regularization* loss encourages smoothness between the pixels. Its popular *self-supervised* variant is

$$P(\sigma_i, \sigma_j) = w_{i,j} \cdot [\sigma_i \neq \sigma_j]$$

where pairwise affinities $w_{ij}$ are based on local intensity edges [6, 17, 22]. Of course, in the context of network training, one should use relaxations of $P$ applicable to (soft) predictions $\sigma_i \in \Delta^K$. Many types of its relaxation [33, 42] were studied in segmentation, e.g. *quadratic* [17], *bi-linear* [36], *total variation* [32, 8], and others [14].

Another unsupervised loss highly relevant for training segmentation networks is the entropy of predictions, which is also known as *decisiveness* [7, 18]

$$\sum_i H(\sigma_i)$$

where $H$ is the Shannon's entropy function. This loss can improve generalization and the quality of representation by moving (deep) features away from the decision boundaries. Widely known in the context of unsupervised or semi-supervised classification, this loss also matters in weakly-supervised segmentation where it is used explicitly or implicitly[1].

Other unsupervised losses (e.g. contrastive), clustering criteria (e.g. K-means), or specialized architectures can be found in weakly-supervised segmentation [39, 31, 20, 9]. However, a lot can be achieved simply by combining the basic losses discussed above

$$L_{ws}(\sigma) \;\; := \;\; -\sum_{i \in S} \ln \sigma_i^{\bar{y}_i} \;+\; \eta \sum_{i \notin S} H(\sigma_i) \;+\; \lambda \sum_{ij \in \mathcal{N}} P(\sigma_i, \sigma_j) \tag{4}$$

which can be optimized directly by gradient descent [36, 38] or using *self-labeling* techniques [26, 28, 27] incorporating optimization of auxiliary *pseudo-labels* as sub-problems.

---

[1]Interestingly, a unary decisiveness-like term is the difference between convex quadratic and *tight*, but non-convex, bi-linear relaxations [33, 27] of the discrete pairwise Potts model.

## 1.2 Soft pseudo-labels: motivation and contributions

We observe that self-labeling with hard pseudo-labels $y_i$, which is discussed in the Appendix A, is inherently limited as such labels can not represent the uncertainty of class estimates at unlabeled pixels $i \in \Omega \backslash S$. Instead, we focus on *soft* pseudo-labels

$$y_i \;=\; (y_i^1, \ldots, y_i^K) \in \Delta^K \tag{5}$$

which are general categorical distributions $p$ over $K$-classes. It is possible that the estimated pseudo-label $y_i$ in (5) could be a one-hot distribution, which is a vertex of $\Delta^K$. In such a case, one can treat $y_i$ as a class index, but we avoid this in the main part of our paper starting Section 2. However, the ground truth labels $\bar{y}_i$ are always hard and we use them either as indices (1) or one-hot distributions (2), as convenient.

Soft pseudo-labels can be found in prior work on weakly-supervised segmentation [25, 41] using the "soft proposal generation". In contrast, we formulate soft self-labeling as a principled optimization methodology where network predictions and soft pseudo-labels are variables in a joint loss, which guarantees convergence of the training procedure. Our pseudo-labels are auxiliary variables for ADM-based [5] splitting of the loss (4) into two simpler optimization sub-problems: one focused on the Potts model over unlabeled pixels, and the other on the network training. While similar to [28], instead of hard, we use soft auxiliary variables for the Potts sub-problem. Our work can be seen as a study of the relaxed Potts sub-problem in the context of weakly-supervised semantic segmentation. The related prior work is focused on discrete solvers fundamentally unable to represent class estimate uncertainty. Our contributions can be summarized as follows:

- convergent *soft self-labeling* framework based on a simple joint self-labeling loss
- systematic evaluation of Potts relaxations and (cross-) entropy terms in our loss
- state-of-the-art in scribble-based semantic segmentation that does not require any modifications of semantic segmentation models and is easy to reproduce
- using the same segmentation model, our self-labeling loss with $3\%$ scribbles may outperform standard supervised cross-entropy loss with full ground truth masks.

## 2 Our soft self-labeling approach

First, we apply ADM splitting [5] to weakly supervised loss (4) to formulate our self-labeling loss (6) incorporating additional soft auxiliary variables, i.e. pseudo-labels (5). It is convenient to introduce pseudo-labels $y_i$ on all pixels in $\Omega$ even though a subset of pixels (seeds) $S \subset \Omega$ have ground truth labels $\bar{y}_i$. We will simply impose a constraint that pseudo-labels and ground truth labels agree on $S$. Thus, we assume the following set of pseudo-labels

$$Y_\Omega := \{y_i \in \Delta^K \,|\, i \in \Omega, \text{ s.t. } y_i = \bar{y}_i \text{ for } i \in S\}.$$

We split the terms in (4) into two groups: one includes NLL and entropy $H$ terms keeping the original prediction variables $\sigma_i$ and the other includes the Potts relaxation $P$ replacing $\sigma_i$ with auxiliary variables $y_i$. This transforms loss (4) into expression

$$-\sum_{i \in S} \ln \sigma_i^{\bar{y}_i} \;+\; \eta \sum_{i \notin S} H(\sigma_i) \;+\; \lambda \sum_{ij \in \mathcal{N}} P(y_i, y_j)$$

equivalent to (4) assuming equality $\sigma_i = y_i$. The standard approximation is to incorporate constraint $\sigma_i \approx y_i$ directly into the loss, e.g. using $KL$-divergence. For simplicity, we use weight $\eta$ for $KL(\sigma_i, y_i)$ to combine it with $H(\sigma_i)$ into a single cross-entropy term

$$-\sum_{i \in S} \ln \sigma_i^{\bar{y}_i} \;+\; \underbrace{\eta \sum_{i \notin S} H(\sigma_i) \;+\; \eta \sum_{i \notin S} KL(\sigma_i, y_i)}_{\eta \sum_{i \notin S} H(\sigma_i, y_i)} \;+\; \lambda \sum_{ij \in \mathcal{N}} P(y_i, y_j)$$

defining joint *self-labeling loss* for both predictions $\sigma_i$ and pseudo-labels $y_i$

$$L_{self}(\sigma, y) \;:=\; -\sum_{i \in S} \ln \sigma_i^{\bar{y}_i} \;+\; \eta \sum_{i \notin S} H(\sigma_i, y_i) \;+\; \lambda \sum_{ij \in \mathcal{N}} P(y_i, y_j) \tag{6}$$

| bi-linear ∼ "graph cut" | quadratic ∼ "random walker" |
|---|---|
| $P_{\text{BL}}(p, q) \ := \ 1 - \ p^\top q$ | $P_{\text{Q}}(p, q) \ := \ \frac{1}{2}\|p - q\|^2$ |

| normalized quadratic | | |
|---|---|---|
| $P_{\text{NQ}}(p, q) \ := \ 1 - \frac{p^\top q}{\|p\|\|q\|}$ | $\equiv$ | $\frac{1}{2}\left\|\frac{p}{\|p\|} - \frac{q}{\|q\|}\right\|^2$ |

Table 1: Second-order Potts relaxations, see Fig.1(a,b,c)

85 approximating the original weakly supervised loss (4).

86 Iterative minimization of this loss w.r.t. predictions $\sigma_i$ (model parameters training) and pseudo-
87 labels $y_i$ effectively breaks the original optimization problem for (4) into two simpler sub-problems,
88 assuming there is a good solver for optimal pseudo-labels. The latter seems plausible since the unary
89 term $H(\sigma_i, y_i)$ is convex for $y_i$ and the Potts relaxations were widely studied in image segmentation
90 for decades.

91 Section 2.1 discusses standard and new relaxations of the Potts model $P$. Section 2.2 discusses several
92 robust variants of cross-entropy $H$ for connecting predictions with uncertain (soft) pseudo-labels $y_i$
93 estimated for unlabeled points $i \in \Omega \backslash S$. Appendix B proposes an efficient general solver for the
94 corresponding pseudo-labeling sub-problems.

## 2.1 Second-order relaxations of the Potts model

96 We focus on second-order relaxations for two reasons. First, to manage the scope of this study.
97 Second, this includes several important baseline cases (see Table 1): *quadratic*, the simplest convex
98 relaxation popularized by the *random walker* algorithm [17], and *bi-linear*, which is non-convex but
99 *tight* [33] w.r.t. the original discrete Potts model. The latter implies that optimizing it over relaxed
100 variables will lead to a solution consistent with a discrete Potts solver, e.g. *graph cut* [6]. On the
101 contrary, the quadratic relaxation will produce a significantly different soft solution. We investigate
102 such soft solutions.

Figure 2 shows two examples illustrating local minima for (a) the bi-linear and (b) quadratic relax-
ations of the Potts loss. In (a) two neighboring pixels attempt to jointly change the common soft
label from $y_i = y_j = (1, 0, 0)$ to $y_i'' = y_j'' = (0, 1, 0)$, which corresponds to a "move" where the
whole object is reclassified from A to B. This move does not violate smoothness within the region
represented by the Potts model. But, the soft intermediate state $y_i' = y_j' = (\frac{1}{2}, \frac{1}{2}, 0)$ will prevent this
move in bi-linear case

$$P_{\text{BL}}(y_i', y_j') = \frac{1}{2} \ > \ 0 = P_{\text{BL}}(y_i, y_j) = P_{\text{BL}}(y_i'', y_j'')$$

while quadratic relaxation assigns zero loss for all states during this move. On the other hand, the
example in Figure 2(b) shows a move problematic for the quadratic relaxation. Two neighboring
pixels have labels $y_i = (1, 0, 0)$ and $y_j = (0, 0, 1)$ corresponding to the boundary of objects A and
C. The second object attempts to change from C to B. This move does not affect the discontinuity
between two pixels, but quadratic relaxation prefers that the second object is stuck in the intermediate
state $y_j' = (0, \frac{1}{2}, \frac{1}{2})$

$$P_{\text{Q}}(y_i, y_j') = \frac{3}{4} \ < \ 1 = P_{\text{Q}}(y_i, y_j) = P_{\text{Q}}(y_i, y_j'')$$

103 while bi-linear relaxation $P_{\text{BL}}(y_i, y_j) = 1$ remains constant as $y_j$ changes.

104 We propose a new relaxation, *normalized quadratic* in Table 1. Normalization leads to equivalence
105 between quadratic and bi-linear formulations combining their benefits. As easy to check, normalized

| collision cross entropy | log-quadratic | |
|---|---|---|
| $P_{\text{CCE}}(p, q) \ := \ -\ln p^\top q$ | $P_{\text{LQ}}(p, q) \ := \ -\ln\left(1 - \frac{\|p - q\|^2}{2}\right)$ | |

| collision divergence | | |
|---|---|---|
| $P_{\text{CD}}(p, q) \ := \ -\ln\frac{p^\top q}{\|p\|\|q\|}$ | $\equiv$ | $-\ln\left(1 - \frac{1}{2}\left\|\frac{p}{\|p\|} - \frac{q}{\|q\|}\right\|^2\right)$ |

Table 2: Log-based Potts relaxations, see Fig.1(d,e,f)

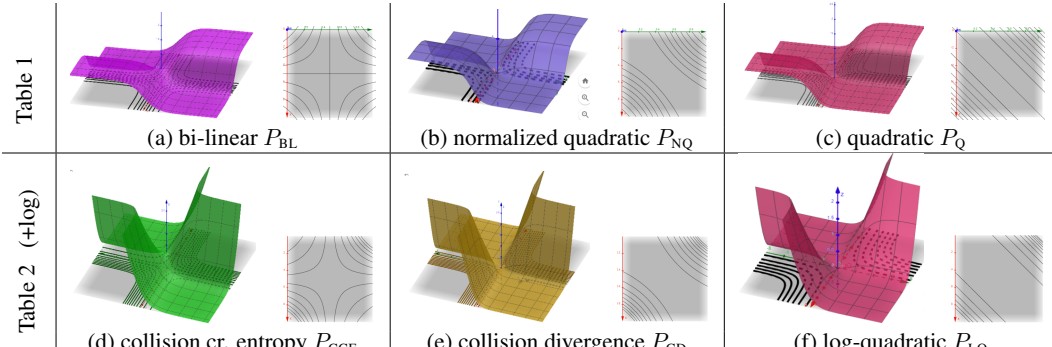

Figure 1: Second-order Potts relaxations in Tables 1 and 2: interaction potentials $P$ for pairs of predictions $(\sigma_i, \sigma_j)$ in (4) or pseudo-labels $(y_i, y_j)$ in (6) are illustrated for $K = 2$ when each prediction $\sigma_i$ or label $y_i$, i.e. distribution in $\Delta^2$, can be represented by a single scalar as $(x, 1 - x)$. The contour maps are iso-levels of $P((x_i, 1 - x_i), (x_j, 1 - x_j))$ over domain $(x_i, x_j) \in [0, 1]^2$. The 3D plots above illustrate the potentials $P$ as functions over pairs of "logits" $(l_i, l_j) \in \mathbb{R}^2$ where each scalar logit $l_i$ defines binary distribution $(x_i, 1 - x_i)$ for $x_i = \frac{1}{1 + e^{-2l_i}} \in [0, 1]$.

quadratic relaxation $P_{\text{NQ}}$ does not have local minima in both examples of Figure 2. Table 2 also proposes "logarithmic" versions of the relaxations in Table 1 composing them with function $-\ln(1 - x)$. As illustrated by Figure 1, the logarithmic versions in (d-f) addresses the "vanishing gradients" evident in (a-c).

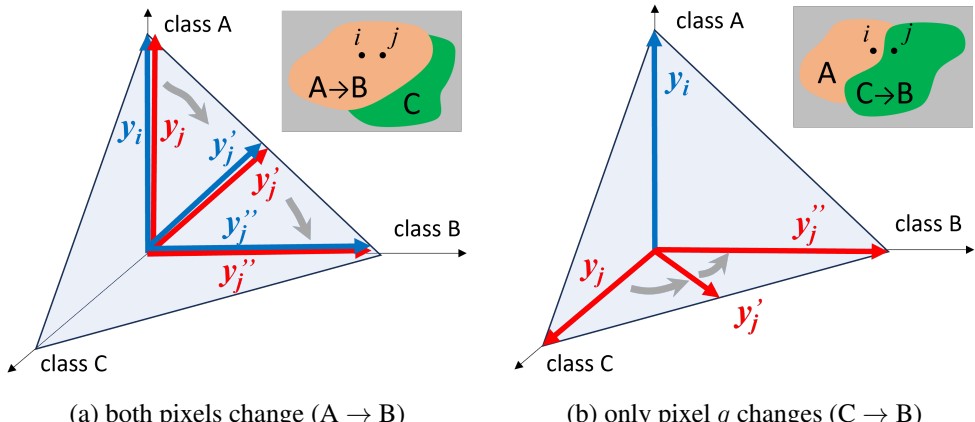

(a) both pixels change (A → B)   (b) only pixel $q$ changes (C → B)

Figure 2: Examples of "moves" for neighboring pixels $\{i, j\} \in \mathcal{N}$. Their (soft) pseudo-labels $y_i$ and $y_j$ are illustrated on the probability simplex $\Delta^K$ for $K = 3$. In (a) both pixels $i$ and $j$ are inside a region/object changing its label from A to B. In (b) pixels $i$ and $j$ are on the boundary between two regions/objects; one is fixed to class A and the other changes from class C to B.

## 2.2 Cross-entropy and soft pseudo-labels

Shannon's cross-entropy $H(y, \sigma)$ is the most common loss for training network predictions $\sigma$ from ground truth labels $y$ in the context of classification, semantic segmentation, etc. However, this loss may not be ideal for applications where the targets $y$ are soft categorical distributions representing various forms of class uncertainty. For example, this paper is focused on scribble-based segmentation where the ground truth is not known for most of the pixels, and the network training is done jointly with estimating *pseudo-labels* $y$ for the unlabeled pixels. In this case, soft labels $y$ are distributions representing class uncertainty. We observe that if such $y$ is used as a target in $H(y, \sigma)$, the network is trained to reproduce the uncertainty, see Figure 3(a). This motivates the discussion of alternative "cross-entropy" functions where the quotes indicate an informal interpretation of this information-theoretic concept. Intuitively, such functions should encourage decisiveness, as well as proximity

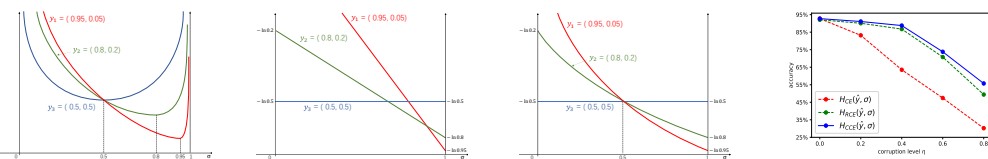

(a) standard $H_{\text{CE}}(y, \sigma)$    (b) reverse $H_{\text{RCE}}(y, \sigma)$    (c) collision $H_{\text{CCE}}(y, \sigma)$    (d) empirical comparison

Figure 3: Illustration of cross-entropy functions: (a) standard (7), (b) reverse (8), and (c) collision (9). (d) shows the empirical comparison on the robustness to label uncertainty. The test uses ResNet-18 architecture on fully-supervised *Natural Scene* dataset [30] where we corrupted some labels. The horizontal axis shows the percentage $\eta$ of training images where the correct ground truth labels were replaced by a random label. All losses trained the model using soft target distributions $\hat{y} = \eta * u + (1 - \eta) * y$ representing the mixture of one-hot distribution $y$ for the observed corrupt label and the uniform distribution $u$, following [29]. The vertical axis shows the test accuracy. Training with the reverse and collision cross-entropy is robust to much higher levels of label uncertainty.

121 between the predictions and pseudo-labels, but avoid mimicking the uncertainty in both directions:
122 from soft pseudo-labels to predictions and vice-versa. We show that the last property can be achieved
123 in a probabilistically principled manner. The following three paragraphs discuss different cross-
124 entropy functions that we study in the context of our self-labeling loss (6).

125 **Standard cross-entropy** provides the obvious baseline for evaluating two alternative versions that
126 follow. For completeness, we include its mathematical definition

$$H_{\text{CE}}(y_i, \sigma_i) \;\; = \;\; H(y_i, \sigma_i) \;\; \equiv \;\; -\sum_k y_i^k \ln \sigma_i^k \tag{7}$$

127 and remind the reader that this loss is primarily used with hard or one-hot labels, in which case it is
128 also equivalent to NLL loss $-\ln \sigma_i^{y_i}$ previously discussed for ground truth labels (3). As mentioned
129 earlier, Figure 3(a) shows that for soft pseudo-labels like $y = (0.5, 0.5)$, it forces predictions to mimic
130 or replicate the uncertainty $\sigma \approx y$. In fact, label $y = (0.5, 0.5)$ just tells that the class is unknown
131 and the network should not be supervised by this point. This problem manifests itself in the poor
132 performance of the standard cross-entropy (7) in our experiment discussed in Figure 3 (d) (red curve).

133 **Reverse cross-entropy** switches the order of the label and prediction in (7)

$$H_{\text{RCE}}(y_i, \sigma_i) \;\; = \;\; H(\sigma_i, y_i) \;\; \equiv \;\; -\sum_k \sigma_i^k \ln y_i^k \tag{8}$$

134 which is not too common. Indeed, Shannon's cross-entropy is not symmetric and the first argument
135 is normally the *target* distribution and the second is the *estimated* distribution. However, in our
136 case, both distributions are estimated and there is no reason not to try the reverse order. It is worth
137 noting that our self-labeling formulation (6) suggests that reverse cross-entropy naturally appears
138 when the ADM approach splits the decisiveness and fairness into separate sub-problems. Moreover,
139 as Figure 3(b) shows, in this case, the network does not mimic uncertain pseudo-labels, e.g. the
140 gradient of the blue line is zero. The results for the reverse cross-entropy in Figure 3 (d) (green)
141 are significantly better than for the standard (red). Unfortunately, now pseudo-labels $y$ mimic the
142 uncertainty in predictions $\sigma$.

143 **Collision cross-entropy** resolves the problem in a principled way. We define it as

$$H_{\text{CCE}}(y_i, \sigma_i) \;\; \equiv \;\; -\ln \sum_k \sigma_i^k y_i^k \;\; \equiv \;\; -\ln \sigma^\top y \tag{9}$$

which is symmetric w.r.t. pseudo-labels and predictions. The dot product $\sigma^\top y$ can be seen as a probability that random variables represented by the distribution $\sigma$, the prediction class $C$, and the distribution $y$, the unknown true class $T$, are equal. Indeed,

$$\Pr(C = T) = \sum_k Pr(C = k) \Pr(T = k) = \sigma^\top y.$$

144 Loss (9) maximizes this "collision" probability rather than the constraint $\sigma = y$. Figure 3(c) shows no
145 mimicking of uncertainty (blue line). However, unlike reverse cross-entropy, this is also valid when $y$

is estimated from uncertain predictions $\sigma$ since (9) is symmetric. This leads to the best performance in Figure 3 (d) (blue). Our extensive experiments are conclusive that collision cross-entropy is the best option for $H$ in self-labeling loss (6).

# 3 Experiments

We conducted comprehensive experiments to demonstrate the choice of each element (cross-entropy, pairwise term, and neighborhood) in the loss and compare our method to the state-of-the-art. In Section 3.1, quantitative results are shown to compare different Potts relaxations. The qualitative examples are shown in Figure 7. Then we compare several cross-entropy terms in Section 3.2. Besides, we also compare our soft self-labeling approach on the nearest and dense neighborhood systems in Section 3.3. We summarized the results in Section 3.4. In the last section, we show that our method achieves the SOTA and even can outperform the fully-supervised method. More details on the dataset, implementation, and additional experiments are given in Appendix C.

## 3.1 Comparison of Potts relaxations

To compare different Potts relaxations under the self-labeling framework, we need to choose one cross-entropy term. Motivated by the properties and empirical results shown in Section 3.2, we use $H_{\text{CCE}}$. The neighborhood system is the nearest neighbors. The quantitative results are in Table 3. First, One can see that the pairwise terms with logarithm are better than those without the logarithm because the logarithm may help with the gradient vanishing problem in softmax operation. Moreover, the logarithm does not like abrupt change across the boundaries, so the transition across the boundaries is smoother (see Figure 7 in the appendix.). Note that it is reasonable to have higher uncertainty around the boundaries. Second, the results prefer the normalized version,

|  | scribble length ratio | | | | |
|---|---|---|---|---|---|
|  | 0 | 0.3 | 0.5 | 0.8 | 1.0 |
| $P_{\text{BL}}$ | 56.42 | 61.74 | 63.81 | 65.73 | 67.24 |
| $P_{\text{NQ}}$ | 59.01 | 65.53 | 67.80 | 70.63 | 71.12 |
| $P_{\text{Q}}$ | 58.92 | 65.34 | 67.81 | 70.43 | 71.05 |
| $P_{\text{CCE}}$ | 56.40 | 61.82 | 63.81 | 65.81 | 67.41 |
| $P_{\text{CD}}$ | 59.04 | 65.52 | 67.84 | 70.93 | 71.22 |
| $P_{\text{LQ}}$ | 59.03 | 65.44 | 67.81 | 70.80 | 71.21 |

Table 3: Comparison of Potts relaxations with self-labeling. mIoUs on validation set are shown here.

which confirms the points made in Figure 2. Third, the simplest quadratic formulation $P_{\text{Q}}$ can be a fairly good starting point to obtain decent results. Additionally, we specifically test $H_{\text{Q}} + P_{\text{Q}}$ due to the existing closed-form solution [1, 17]. Since the pseudo-labels generated from this formula tend to be overly soft, we explicitly add entropy terms during the training of network parameters and the mIoU goes up to 68.97% from 67.8%.

## 3.2 Comparison of cross-entropy terms

In this section, we compare different cross-entropy terms while fixing the pairwise term to $P_{\text{Q}}$ due to its simplicity and using the nearest neighborhood system. The results are shown in Figure 4. One can see that $H_{\text{CCE}}$ performs the best consistently across different supervision levels, i.e. scribble lengths. Both $H_{\text{CCE}}$ and $H_{\text{RCE}}$ are consistently better than standard $H_{\text{CE}}$ with a noticeable margin because they are more robust, as explained in Section 2.2, to the uncertainty in soft pseudo-labels when optimizing network parameters. We also test the performance of using $H_{\text{CCE}} + P_{\text{Q}}$ with hard pseudo-labels obtained via the $argmax$ operation on the soft ones. The mIoU on the validation set is 69.8% under the full scribble-length supervision.

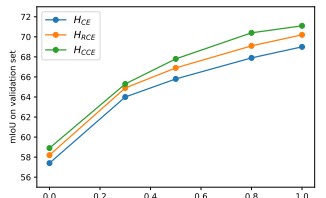

Figure 4: Comparison of cross-entropy terms.

## 3.3 Comparison of neighborhood systems

Until now, we only used the four nearest neighbors for the pairwise term. In this section, we also use the dense neighborhood and compare the results under the self-labeling framework.

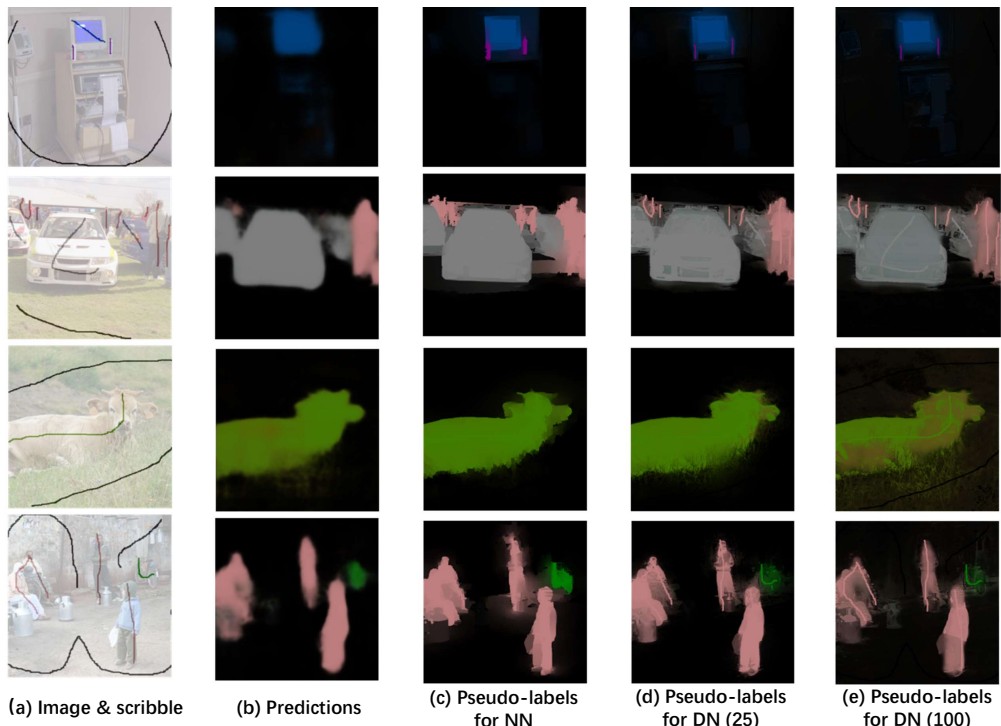

| (a) Image & scribble | (b) Predictions | (c) Pseudo-labels for NN | (d) Pseudo-labels for DN (25) | (e) Pseudo-labels for DN (100) |

Figure 5: Pseudo-labels generated from given network predictions using different neighborhoods.

Firstly, to optimize the pseudo-labels for the dense neighborhood, we still use the gradient descent technique as detailed in Appendix B. The gradient computation employs the bilateral filtering technique following [35]. For the pairwise term, we use $P_Q$. The cross-entropy term is $H_{CCE}$. Note that the bilateral filtering technique only supports quadratic pairwise terms, i.e. $P_{BL}$ and $P_Q$. Since $P_{BL}$ leads to hard solutions, $P_Q$ is the only practical choice for soft self-labeling. We obtained 71.1% mIoU on nearest neighbors while only getting 67.9% on dense neighborhoods (bandwidth is 100). Some qualitative results are shown in Figure 5. Clearly from this figure one can see that a larger neighborhood size induces lower-quality pseudo-labels. A possible explanation is that the Potts model gets closer to cardinality/volume potentials when the neighborhood size becomes larger [37]. The nearest neighborhood is better for edge alignment and thus produces cleaner results.

## 3.4 Soft self-labeling vs. hard self-labeling vs. gradient descent

In this section, we give a summary in Table 4 as to what is the best framework for the WSSS based on losses regularized by the Potts model. Firstly, to directly optimize the network parameters via stochastic gradient descent on the regularized loss, one needs a larger neighborhood size. One possible explanation is that a larger neighborhood size induces a smoother Potts model and it helps the gradient descent [28]. However, larger neighborhood size is not preferred in the self-labeling framework. If we use Potts model on nearest neighborhoods, the self-labeling optimization should be applied and one should use

|  |  | $\mathcal{N}$ | |
|---|---|---|---|
|  |  | NN | DN |
| GD |  | 67.0 | 69.5* [36] |
| SL | hard | 69.6* [27] | 63.1 [26] |
|  | soft | **71.1** | 67.9 |

Table 4: Summary of comparisons. "*" stands for the reproduced results from their code repository.

soft pseudo-labels instead of hard ones. Note that with proper optimization the advantage of the Potts model on small neighborhood size can show up. In Figure 6, we also compare these approaches across different scribble lengths.

## 3.5 Comparison to SOTA

In this section, we use a different network architecture, ResNet101, to fairly compare our method with the current state-of-the-art. We only compare the results before applying any post-processing steps. The results are shown in Table 5. Note that our results can outperform the fully-supervised method when using 12 as the batch size. We also observe that a larger batch size usually improves the results quite a lot. Our results with 12 batch size can outperform several SOTA methods which use 16 batch size.

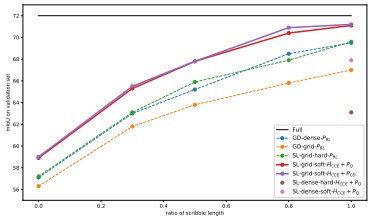

Figure 6: Comparison of different methods using Potts relaxations. The architecture is DeeplabV3+ with the backbone MobileNetV2.

| Method | Architecture | Batchsize | Optimization | | | $\mathcal{N}$ | mIoU |
|---|---|---|---|---|---|---|---|
| | | | GD | SL | | | |
| | | | | hard | soft | | |
| **Full supervision** | | | | | | | |
| Deeplab* [12] | V3+ | 16 | ✓ | - | - | - | 78.9 |
| Deeplab* [12] | V3+ | 12 | ✓ | - | - | - | 76.6 |
| Deeplab [11] | V2 | 12 | ✓ | - | - | - | 75.6 |
| **Scribble supervision** | | | | | | | |
| *Architectural modification* | | | | | | | |
| BPG [39] | V2 | 10 | ✓ | - | - | - | 73.2 |
| URSS [31] | V2 | 16 | ✓ | - | - | - | 74.6 |
| SPML [20] | V2 | 16 | ✓ | - | - | - | 74.2 |
| PSI [41] | V3+ | - | - | - | ✓ | - | 74.9 |
| SEMINAR [9] | V3+ | 12 | ✓ | - | - | - | 76.2 |
| TEL [25] | V3+ | 16 | - | - | ✓ | - | 77.1 |
| *Loss modification - Potts relaxations* | | | | | | | |
| ScribbleSup [26] | VGG16(V2) | 8 | - | ✓ | - | DN | 63.1 |
| DenseCRF loss* [36] | V3+ | 12 | ✓ | - | - | DN | 75.8 |
| GridCRF loss* [27] | V3+ | 12 | - | ✓ | - | NN | 75.6 |
| NonlocalCRF loss* [38] | V3+ | 12 | ✓ | - | - | SN | 75.7 |
| $\mathbf{H}_{\text{CCE}} + \mathbf{P}_{\text{Q}}$ | V3+ | 12 | - | - | ✓ | NN | 77.5 |
| $\mathbf{H}_{\text{CCE}} + \mathbf{P}_{\text{CD}}$ | V3+ | 12 | - | - | ✓ | NN | **77.7** |
| $\mathbf{H}_{\text{CCE}} + \mathbf{P}_{\text{CD}}$ (no pretrain) | V3+ | 12 | - | - | ✓ | NN | 76.7 |
| $\mathbf{H}_{\text{CCE}} + \mathbf{P}_{\text{CD}}$ | V3+ | 16 | - | - | ✓ | NN | **78.1** |
| $\mathbf{H}_{\text{CCE}} + \mathbf{P}_{\text{CD}}$ (no pretrain) | V3+ | 16 | - | - | ✓ | NN | 77.6 |

Table 5: Comparison to SOTA methods (without CRF postprocessing) on scribble-supervised segmentation. The numbers are mIoU on the validation dataset of Pascal VOC 2012 and use full-length scribble. The backbone is ResNet101 unless stated otherwise. V2: deeplabV2. V3+: deeplabV3+. $\mathcal{N}$: neighborhood. "$*$": reproduced results. GD: gradient descent. SL: self-labeling. "no pretrain" means the segmentation network is not pretrained using cross-entropy on scribbles.

## 4 Conclusions

This paper proposed a convergent soft self-labeling framework based on a simple well-motivated loss (6) for joint optimization of network predictions and soft *pseudo-labels*. The latter were motivated as auxiliary optimization variables simplifying optimization of weakly-supervised loss (4). Our systematic evaluation of the cross-entropy and the Potts terms in self-labeling loss (6) provides clear recommendations based on the discussed conceptual advantages empirically confirmed by our experiments. Specifically, our work recommends the collision cross-entropy, log-quadratic Potts relaxations, and the earest-neighbor neighborhood. They achieve the best result that may even outperform the fully-supervised method with full pixel-precise masks. Our method does not require any modifications of the semantic segmentation models and it is easy to reproduce. Our general framework and empirical findings can be useful for other weakly-supervised segmentation problems (boxes, class tags, etc.).

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

# A  Self-labeling and hard pseudo-labels

One argument motivating self-labeling approaches to weakly-supervised segmentation comes from well-known limitations of gradient descent when optimizing the Potts relaxatons, e.g. [28]. But even when using convex Potts relaxations [17, 32, 8], they are combined with the concave entropy term in (4) making their optimization challenging.

Typical self-labeling methods, including one of the first works on scribble-based semantic segmentation [26], introduce a sub-problem focused on the estimation of *pseudo-labels* over unlabeled points, separately from the network training by such labels. Pseudo-labeling is typically done by optimization algorithms or heuristics balancing unsupervised or self-supervised criteria, e.g. the Potts, and proximity to current predictions. Then, network fine-tuning from pseudo-labels and pseudo-labeling steps are iterated.

We denote pseudo-labels $y_i$ slightly differently from the ground truth labels $\bar{y}_i$ by omitting the bar. It is important to distinguish them since the ground truth labels $\bar{y}_i$ for $i \in S$ are given, while the pseudo-labels $y_i$ for $i \in \Omega \backslash S$ are estimated. The majority of existing self-labeling methods [26, 2, 28, 3, 24, 27, 40] estimate *hard* pseudo-labels, which could be equivalently represented either by class indices

$$y_i \in \{1, \ldots, K\} \tag{10}$$

or by the corresponding one-hot categorical distributions

$$y_i \equiv (y_i^1, \ldots, y_i^K) \in \Delta_{0,1}^K \qquad \text{for} \quad y_i^k := [k = y_i] \in \{0, 1\} \tag{11}$$

analogously with the hard ground truth labels in (1) and (2). In part, hard pseudo-labels are motivated by the network training where the default is NLL loss (3) assuming discrete labels. Besides, there are powerful discrete solvers for the Potts model [6, 32, 8]. We discuss the potential advantages of soft pseudo-labels in the next Section 1.2.

**Joint loss vs "proposal generation"**: The majority of self-labeling approaches can be divided into two groups. One group designs pseudo-labeling and the network training sup-problems that are not formally related, e.g. [26, 25, 41]. While pseudo-labeling typically depends on the current network predictions and the network fine-tuning uses such pseudo-labels, the lack of a formal relation between these sub-problems implies that iterating such steps does not guarantee any form of convergence. Such methods are often referred to as *proposal generation* heuristics.

Alternatively, the pseudo-labeling sub-problem and the network training sub-problem can be formally derived from a weakly-supervised loss like (4), e.g. by ADM *splitting* [28] or as high-order *trust-region* method [27]. Such methods often formulate a *joint loss* function w.r.t network predictions and pseudo-labels and iteratively optimize it in a convergent manner that is guaranteed to decrease the loss. We consider this group of self-labeling methods as better motivated, more principled, and numerically safer.

# B  Optimization Algorithm

In this section, we will focus on the optimization of (6) in steps iterating optimization of $y$ and $\sigma$. The network parameters are optimized by standard stochastic gradient descent in all our experiments. Pseudo-labels are also estimated online using a mini-batch. To solve $y$ at given $\sigma$, it is a large-scale constrained convex problem. While there are existing general solvers to find global optima, such as projected gradient descent, it is often too slow for practical usage. Instead, we reformulate our problem to avoid the simplex constraints so that we can use standard gradient descent in PyTorch library accelerated by GPU. Specifically, instead of directly optimizing $y$, we optimize a set of new variables $\{l_i \in \mathbb{R}^K, i \in \Omega\}$ where $y_i$ is computed by $softmax(l_i)$. Now, the simplex constraint on $y$ will be automatically satisfied. Note that the hard constraints on scribble regions still need to be considered because the interaction with unlabeled regions through pairwise terms will influence the optimization process. Inspired by [44], we can reset $softmax(l_i)$ where $i \in S$ back to the ground truth at the beginning of each step of the gradient descent.

However, the original convex problem now becomes non-convex due to the Softmax operation. Thus, initialization is important to help find better local minima or even the global optima. Empirically, we observed that the network output logit can be a fairly good initialization. The quantitative comparison

(a) Image, GT & input

(b) Pseudo-labels using different Potts relaxation

Figure 7: Illustration of the difference among Potts relaxations. The visualization of soft pseudo-labels uses the convex combination of RGB colors for each class weighted by pseudo-label itself.

uses a special quadratic formulation where closed-form solution and efficient solver [1, 17] exist. We compute the standard soft Jaccard index for the pseudo-labels between the solutions given by our solver and the global optima. The soft Jaccard index is 99.2% on average over 100 images. Furthermore, our experimental results for all other formulations in Figure 7, 5, and Section 3 confirm the effectiveness of our optimization solver. In all experiments, the number of gradient descent steps for solving $y$ is 200 and the corresponding learning rate is 0.075. To test the robustness of the number of steps here, we decreased 200 to 100 and the mIoU on the validation set just dropped from 71.05 by 0.72. This indicates that we can significantly accelerate the training without much sacrifice of accuracy. When using 200 steps, the total time for the training will be about 3 times longer than the SGD with dense Potts [36].

## C   Experimental settings

**Dataset and evaluation**   We mainly use the standard PASCAL VOC 2012 dataset [16] and scribble-based annotations for supervision [26]. The dataset contains 21 classes including background. Following the common practice [10, 35, 36], we use the augmented version which has 10,582 training images and 1449 images for validation. We employ the standard mean Intersection-over-Union (mIoU) on validation set as the evaluation metric. We also test our method on two additional datasets in Section 3.5. One is Cityscapes [13] which is built for urban scenes and consists of 2975 and 500 fine-labeled images for training and validation. There are 19 out of 30 annotated classes for semantic segmentation. The other one is ADE20k [43] which has 150 fine-grained classes. There are 20210 and 2000, images for training and validation. Instead of scribble-based supervision, we followed [25] to use the block-wise annotation as a form of weak supervision.

**Implementation details**   We adpoted DeepLabv3+ [12] framework with two backbones, ResNet101 [19] and MobileNetV2 [34]. We use ResNet101 in Section 3.5, and use MobileNetV2 in other sections for efficiency. All backbone networks (ResNet-101 and MobileNetV2) are pre-trained on Imagenet [15]. Unless stated explicitly, we use batch 12 as the default across all the experiments. Following [35], we adopt two-stage training, unless otherwise stated, where only the cross-entropy loss on scribbles is used in the first stage. The optimizer for network parameters is SGD. The learning rate is scheduled by a polynomial decay with a power of 0.9. Initial learning is set to 0.007 in the first stage and 0.0007 in the second phase. 60 epochs are used to train the model with different losses where hyperparameters are tuned separately for them. For our best result, we use $\eta = 0.3, \lambda = 6,$

$H_{\text{CCE}}$ and $P_{\text{CD}}$. The color intensity bandwidth in the Potts model is set to 9 across all the experiments on Pascal VOC 2012 and 3 for Cityscapes and ADE20k datasets.

| Method | Architecture | Cityscapes | ADE20k |
|---|---|---|---|
| **Full supervision** | | | |
| Deeplab [12] | V3+ | 80.2 | 44.6 |
| **Block-scribble supervision** | | | |
| DenseCRF loss [36] | V3+ | 69.3 | 37.4 |
| GridCRF loss* [27] | V3+ | 69.5 | 37.7 |
| TEL [25] | V3+ | 71.5 | 39.2 |
| $\mathbf{H}_{\text{CCE}} + \mathbf{P}_{\text{CD}}$ | V3+ | 72.4 | 39.7 |

Table 6: Comparison to SOTA methods (without CRF postprocessing) on segmentation with block-scribble supervision. The numbers are mIoU on the validation dataset of cityscapes [13] and ADE20k [43] and use $50\%$ of full annotations for supervision following [25]. The backbone is ResNet101. "$*$": reproduced results. All methods are trained in a single-stage fashion.

