# OpenReview forum: "Potts Relaxations and Soft Self-labeling for Weakly-Supervised Segmentation"
_NeurIPS.cc/2024/Conference — Submitted to NeurIPS 2024_

### Official Review · Reviewer_dJES · 2024-06-15

**Soundness:** 3
**Presentation:** 1
**Contribution:** 1
**Rating:** 4
**Confidence:** 5

**Summary:**

This paper proposes a method to improve weakly-supervised semantic segmentation using sparse annotations. The authors introduce a Potts relaxation method, which is an extension of the traditional CRF-like methods. The experiments are conducted on the PASCAL dataset.

**Strengths:**

Pros.

1.	The proposed method appears to be technically sound.

2.	The authors provide detailed explanations of their method.

**Weaknesses:**

Weakness:

1.	The comparisons with other methods are not comprehensive. Recent advances [1,2,3] in sparsely annotated semantic segmentation are not discussed or compared. Point-supervised is more challenging but it is ignored. The survey in the paper is limited, making it difficult to understand the contributions.

2.	The use of a small dataset like PASCAL may not demonstrate the superiority of the proposed method. Results from larger datasets like Cityscapes and ADE20K (Table 6) should be emphasized and compared with state-of-the-art methods.

3.	Self-labeling is also well employed in many weakly-supervised methods that use image-tags labels. It seems that your method can also applied to them, am I right? If so, experiments on the COCO dataset would be important.

4.	DeepLab is an old-fashioned network architecture. The authors should prove that Potts relaxation can also work on Vision Transformer since ViT has demonstrated the SOTA performances in both fully- and weakly-supervised semantic segmentation. If using ViT without Potts can already get good performances, the relaxation may be not important.

5.	In recent fully- and weakly-supervised semantic segmentation works, CRF-like methods have been discarded due to their computation cost. The efficiency of Potts relaxation (FLOPs) is not evaluated in your paper, which is very important.

6.	With complex formulation, on the very toy dataset PASCAL, the mIoU only increases by 1% in Table 5 (77.1 to 78.1), while the increased computation cost has not yet been evaluated.

7.	As stated in the abstract “ … can outperform full pixel-precise supervision on PASCAL”, which is not convincing to me. The fully-supervised performance should be considered as the upper bound of weakly-supervised learning. Such results may be caused by unfair settings.

8.	The importance of relaxation should be introduced in the abstract since it is your main contribution. In Section 2.1, the two claimed reasons are not intuitive to me. “Manage the scope of this study” seems not a strong motivation.

9.	With the development of ViT and vision pre-training, the improvements of CRF-based self-labeling become marginal. Thus, I think it is important to evaluate ViT and vision-pretraining (DINO[4]) backbone with your relaxation. I am wondering whether Potts can improve performances beyond these strong backbones.

Overall,  I think it would be better if the authors could conduct more comprehensive experiment comparisons and related work discussions. The motivation for relaxation should be introduced in the beginning.

Refers:

1.	Sparsely Annotated Semantic Segmentation With Adaptive Gaussian Mixtures. CVPR 23

2.	Label-efficient Segmentation via Affinity Propagation. NIPS 23

3.	Modeling the Label Distributions for Weakly-Supervised Semantic Segmentation. Arxiv 24

4.	DINOv2: Learning Robust Visual Features without Supervision. TMLR 24

5. CC4S: Encouraging Certainty and Consistency in Scribble-Supervised Semantic Segmentation. TPAMI 24

**Questions:**

Questions:

Please refer to the weakness.

Suggestion:

1. Add more comprehensive comparisons and discussions of related works.
2. Some important experiments and evaluations could be considered
3. The motivation for relaxation could be introduced in the abstract.

**Limitations:**

Yes.

---

> ### Author Rebuttal · Authors · 2024-08-05
>
> **1. The comparisons with other methods are not comprehensive**\
> Compared to the scribble-supervision results in [1] (76.4%), [2] (76.6%), and [3] (77.5%), see citations by the reviewer, our result (78.1%, see Table 5) using the same architecture (Deeplab + ResNet101 backbone) is better. We will include these extra results into our Table 5 and discuss the relation to these works. E.g. papers [1] and [3] regularizes the features inside the segments, while our Potts model regularizes the boundaries between the segments, just like the "region" and "boundary" terms in the Chan-Vese or Mumford-Shah regularization models for segmentation. The loss defined in [2] is addressed with an optimization method that does not guarantee convergence.  As for [5], our paper already includes a comparison with their conference version (URSS).
>
> While we are glad to add these references into our paper, we would like to emphasize that our work is focused on studying the properties of different relaxations of the standard Potts model (well-known in segmentation since 80's), as well as different cross-entropy terms, in the general context of **soft** self-labeling. Even our current experimental results are comprehensive enough to support our conclusions about the key technical contributions of the paper.
>
> **1 (cont.) Point-supervised is more challenging but it is ignored.**\
> We do have results for point supervision in Figure 6. Even though our method degrades with lower scribble length ratios, it still performs much better than other methods consistently across different scribble length ratios. Segmentation with point supervision and image tag supervision are both more challenging problems and much more work needs to be done to develop principled methods for them.
>
> **2.  larger datasets like Cityscapes and ADE20K (Table 6) should be emphasized and compared with state-of-the-art methods**\
> The Pascal dataset is a standard benchmark for WSSS (based on scribbles). We've already spent much room discussing the properties of our losses. Due to the space limitation, we move the results on other datasets to the appendix and we leave the comprehensive evaluation on more different tasks and more datasets to the future work.
>
> **3. experiments using image tag supervision on the COCO dataset would be important.**\
> We agree that our method is general enough to be applied to any weakly-supervised segmentation task. However, our goal is to study the properties of the proposed losses instead of exhaustive experiments on all tasks. Also, note that for the tag-supervised segmentation, the loss on scribbles must be replaced by losses specific to tag-supervision, which tend to include a multitude of complex terms. Moreover, current competitive tag-based methods are multi-stage and use specialized architectures. It is hard to single out the effect of the Potts model in this case. Scribble-based supervision significantly simplifies the analysis of the Potts model. Yet, our clear findings are useful for future work with any supervision.
>
> **4. ViT has demonstrated SOTA in fully- and weakly-supervised SS... prove that Potts relaxation can also work on ViT**\
> Most prior WSSS papers **based on scribbles** use DeepLab and Resnet backbones. Even recent papers [1,2] from 2023 provided by the reviewer use these architectures and omit ViT. Even if we tested ViT, we wouldn't have many prior works to compare to. The arXiv [3] posted in 2024, just two months before NeurIPS deadline, does include results on ViT, but they do show that the results are significantly improved by CRF postprocessing, which contradicts the reviewer's guess that CRF (Potts model and its relaxations) is not important for ViT.
>
> A relative comparison of our different Potts relaxations in the context of ViT is possible. However, we do not see any technical reasons to expect significant differences compared to relative results in (e.g. Table 3), which are consistent with their theoretical properties shown in our paper. We leave such evaluation for future work.
>
> **5. The efficiency of Potts relaxation (FLOPs) is not evaluated in your paper**\
> We provide the computational cost compared to the method [36] on line 428. In general, our computational costs are on par with or better than related self-labeling methods for WSSS, e.g. [26, 27].
>
> **6. With complex formulation...**\
> Our self-labeling formulation is well-defined and very simple. It only consists of three terms: unary term on scribbles, unary term linking the prediction and pseudo-labels, and pairwise relaxation terms. These three terms are standard, e.g. [28]. One of our main contributions is the comprehensive study of the new variations of the last two terms.
>
> **7. The results that the proposed method can outperform full pixel-precise supervision on PASCAL is not convincing**\
> The pixel-wise annotation is very human labor intensive, and it is more prone to labeling errors than using the scribbles. Indeed, we observed that there are many wrong labels in the original Pascal dataset annotations. This could be the reason our method outperforms the full pixel-precise supervision one.
>
> **9. With the development of ViT and vision pre-training, the improvements of CRF-based self-labeling become marginal**\
> We do not see any evidence supporting this claim. Why should it be marginal? On the contrary, the recent arXiv provided by the reviewer [3] clearly shows that CRF (and the Potts model) is highly relevant for ViT,  as already discussed in point 4. Also, we find it strange that DINO paper [4] did not try standard general CRF losses, which are probably the simplest and the most common unsupervised losses for segmentation. We can speculate that their results may improve even with a standard bi-linear relaxation [36]. However, we do not think it is our responsibility to prove the Potts model for all possible unsupervised or weakly supervised applications. This model is already well-estanlished in the vision community since 1980s.

---

> > ### Comment · Reviewer_dJES · 2024-08-07
> > **Response to Authors from Reviewer dJES**
> >
> > Thank you for your response. I have thoroughly reviewed all of your responses and the comments from other reviewers. While I can accept some of your responses, there are still some concerns that have not been adequately addressed.
> >
> > 1. There are several works in the field of WSSS that have utilized ViT (not just [3]). In reference to my fourth comment, I suggest using pure ViT as a baseline and then comparing it with (ViT + Potts) to assess whether Potts can enhance ViT. If Potts does not lead to improvements in ViT, its contribution may be marginal, considering ViT's superior strength compared to deeplab.
> >
> > 2.  Line 428 only mentions the training time, which can be influenced by various factors. In my fifth comment, I emphasize efficiency, and evaluating FLOPs should only take a few minutes.
> >
> > 3. The essence of my sixth comment lies in the marginal improvements, yet the response shifts the focus to formulation complexity and overlooks this aspect. I maintain that the enhancements remain marginal, as indicated by your response stating that the improvements are less than 1%.
> >
> > 4. Your explanation regarding the point-supervised experiments does not fully convince me. Identifying your point-supervised experiments in Fig. 6 is challenging, and I am uncertain why Potts does not perform well on points. Considering that TEL conducted experiments on both points and scribbles, which serve as the primary comparison objects in the paper.
> >
> > 5. I did not find a response to point 8; could this be an oversight?
> >
> > I am inclined to accept responses 2, 3, 7, and 9 partially. I sincerely appreciate your detailed responses. In consideration of the comments from other reviewers, I am willing to adjust my ratings if the remaining concerns are addressed satisfactorily.

---

> ### Author Response · Authors · 2024-08-12
> **Author's response to the reviewer points 1 and 2**
>
> > 1. There are several works in the field of WSSS that have utilized ViT (not just [3]). In reference to my fourth comment, I suggest using pure ViT as a baseline and then comparing it with (ViT + Potts) to assess whether Potts can enhance ViT. **If Potts does not lead to improvements in ViT, its contribution may be marginal**, considering ViT's superior strength compared to deeplab.
>
> Based on your original review, as well as comments from jiv8 and other reviewers, we started running some ViT experiments. By now we obtained the following results on the ViT backbone (vit_base_patch16_224) on PASCAL with full scribbles. We hope it helps to confirm that Potts matters for WSSS with any backbones, including ViT. The results below show that the Potts model matters with our approach using this loss directly during training, and it matters for [3] using dense CRF (also a version of Potts) as post-processing.
>
> **batch size 12**:
>
> only partial CE  =>  **74.61%** mIOU (**the baseline you suggested**)
>
> partial CE + Log Div Potts +  CCE => **80.80%** mIOU   (**our**)
>
> **batch size 16**:
>
> only partial CE   =>  **75.10%** mIoU  (**the baseline you suggested**)
>
> partial CE + Log Div Potts + CCE => **80.94%** mIOU   (**our**)
>
> **arXiv [3], March 2024 (no batch size reported)**:
>
> =>  **78.7%** mIoU  (**without CRF postprocessing**)
>
> =>   **80.3%** mIoU  (**with CRF post-processing**)
>
> We will be happy to add these numbers to the final version of Table 5 and discuss the importance of the Potts model across architectures, including ViT. Our final numbers for ViT (with batches 12 and 16) might be higher than the above as we might find better tuning (learning rate, number of epochs, etc) when we have a bit more time.
>
> Another related general observation about the numbers above. It is natural to expect that direct integration of the Potts model as a loss for standard end-to-end training may be a simpler and more stable approach than using the Potts model for postprocessing, as in [3]. The latter requires independent tuning of multiple stages.
>
> > 2. Line 428 only mentions the training time, which can be influenced by various factors. In my fifth comment, I emphasize efficiency, and evaluating FLOPs should only take a few minutes.
>
> We find that FLOPS are common for papers focused on architecture. But, based on our experience, it is not common for WSSS papers, perhaps because they typically focus more on unsupervised or weakly supervised losses. Thus, there is a very limited base for comparison in prior work. One exception for WSSS is a recent arXiv [3] from March 2024 brought up by the reviewer.
>
> Moreover, it is unclear how useful such a comparison would be since one can report FLOPs only for one iteration, while the number of iterations may very significantly between the algorithms. For some algorithms there are also post-processing steps, e.g.  [3],  that adds different FLOPs for an unknown number of such steps. Thus, we doubt the usefulness of reporting FLOPs.
>
> In any case, in a good faith effort to address the reviewer’s question, we did compute FLOPS for one iteration of our self-labeling method that combines a forward pass for the network (for deeplab from calflops library) and 200 steps of gradience descent for our pseudo-labeling estimation (we use at each iteration). We could not find FLOPs for the deeplab backpropagation step (for ViT we do not have even the forward numbers).
>
> **185.96** + **59.77** GFLOPs (513 * 513 input size, resnet101 backbone, deeplabV3+)
>
> One useful thing these two numbers show is that the complexity is dominated by network training, not pseudo-label estimation. In any case, why do you think it is important to include these numbers in the paper and what should we compare them with? The numbers in [3] are for a different architecture, making them hard to compare.

---

> ### Author Response · Authors · 2024-08-12
> **Author's response to the reviewer points 3 and 4**
>
> >3. The essence of my sixth comment lies in the marginal improvements, yet the response shifts the focus to formulation complexity and overlooks this aspect. I maintain that the enhancements remain marginal, as indicated by your response stating that the improvements are less than 1%.
>
> We did not realize that you might not be able to see our response to a similar point by Pwad: “Is 1% a meaningful improvement or just noise?"  Below is a copy of our response to this reviewer. We hope it helps.
>
> **About variation**: While we did not properly collect the variations over multiple runs (the tests are expensive even without this), our informal observations are that this variation is very low (below 1%). Also, it is standard in WSSS literature (and prior works we cite) to repost the best run.
>
> **About improvements wrt SOTA**: Table 5 includes many prior scribble segmentation methods, including those that design specialized complex architectures, see the "architectural modifications" block. It makes the most sense to directly compare our results only with methods using standard architectural backbones (the last block in Table 5) since this constitutes a fair comparison of different loss functions for WSSS, which is the focus of our study. For example, one can easily use such general losses, including ours, to build complex systems or specialized architectures, but this is not the focus of our work studying the basic conceptual properties of a large general class of Potts relaxations.
>
> Indeed, according to Table 5, our method with standard V3+ backbone (16 batches) outperforms the method in [25] modifying V3+ architecture (also 16 batches) only by 1%, which may or may not be significant. However, it may not be a fair comparison since [25] designs a more complex 2-branch architecture. Moreover, their approach has some technical flaws as their training is not guaranteed to converge (their procedurally-defined iterative method does not have a clearly defined self-labeling loss). Such ad-hoc methods typically do not generalize well to datasets other than those for which they were designed (e.g. Pascal in this case).
>
> In any case, it makes more sense to compare our results mainly within the last block where we collected many 12-batch results on V3+ from comparable prior work studying loss functions on standard architectures. We consistently outperform those by at least 2% or more only by using new loss functions and a stronger well-defined optimization algorithm. In this 12-batch scenario we even **outperform the full supervision by 1%**. One of such experiments may or may not be significant, but the consistency of our improvements matters particularly because they come only from simple general loss functions that anyone can use in any system or architecture (simple or complex).
>
>
> >4. Your explanation regarding the point-supervised experiments does not fully convince me. Identifying your point-supervised experiments in Fig. 6 is challenging, and I am uncertain why Potts does not perform well on points. Considering that TEL conducted experiments on both points and scribbles, which serve as the primary comparison objects in the paper.
>
> The point supervision in Fig 6 and Fig 4 is the leftmost point that is marked 0%, but it means points only (which mathematically corresponds to scribble area of measure zero). We can emphasize this important information. We missed it because this is standard in scribble supervision, but we should state this.
>
> Points (or very short scribbles in general) are a problem for the original (discrete) Potts due to the implicit “shrinkage” bias (as it minimizes boundary length). This may explain why the 100% scribbles in the standard database are pretty long. In the context of Potts relaxation, our paper discusses that this is the issue for the **tight** relaxations, but non-tight relaxations (e.g. quadratic) do not have this bias - they have other biases. These various biases are discussed in Sec .2.1 and supported by Figs 1 and 2.
>
> TEL experiments in Table 5 are only for 100% scribbles, which does not show the full picture as the results for varying scribble lengths in Fig 6 and Fig 4. We do not see a similar evaluation in [25].

---

> ### Author Response · Authors · 2024-08-12
> **Author's response to the reviewer point 5 (the missed point 8 of the original review)**
>
> >5. Missing response to earlier point 8: “the importance of relaxation should be introduced in the abstract since it is your main contribution. In Section 2.1, the two claimed reasons are not intuitive to me. “Manage the scope of this study” seems not a strong motivation.”
>
> Indeed, we might have overlooked this somehow.
>
> The systematic study of CRF relaxations is stated in the abstract (the sentence stating our contributions, see line 8). We are happy to stress their importance even more.
>
> We can clarify the meaning of “managing the scope of this study.” Note that there are infinitely (uncountably) many ways to relax the standard (discrete) Potts model. Even within the domain of polynomial relaxations, one can choose a polynomial order corresponding to any natural number (we use 2). There are also many different relaxations using the same order polynomials. Moreover, relaxations do not have to be polynomial. Any (convex) combination of different relaxations is also a valid relaxation.
>
> We had to focus on something and second-order polynomials are the simplest form of relaxation (there are no linear relaxations since one can not fit a plane to 4 values P(0.0), P(0,1), P(1,0) and P(1,1) defining the discrete Potts model). Moreover, bilinear and quadratic relaxations are probably the most standard relaxations in the optimization of pairwise CRF terms, but were not systematically evaluated/compared in WSSS. We discuss their properties in Sec 2.1, many of which are well-known. In Sec 2.1 we use this discussion of limitations and biases to motivate new "second-order" relaxations addressing these limitations.

---

> > ### Comment · Reviewer_dJES · 2024-08-13
> > **Response from reviewer dJES**
> >
> > Apologies for the delay in my response. I truly appreciate the efforts made by the authors, and I am pleased to see that they have incorporated some significant experiments to address the concerns raised. Finally, I would like to improve my rating to 'borderline reject" due to some remaining concerns:
> >
> > - In the response, the 1% improvement appears to lack significance, with minimal variation or none at all. This indicates that the novelty compared to previous works is not substantial. Additionally, a 1% enhancement on the Pascal VOC dataset does not represent a significant advancement, especially considering that it is not a particularly challenging dataset.
> > - Upon revisiting TEL [25*], I noted that they have included point-supervised experiments (Table 1). The limitation of Potts not functioning effectively on points is a notable drawback.
> > - Furthermore, as highlighted by the authors, the superior performance of scribble-supervised learning compared to fully-supervised learning implies that scribbles may not be as "weak" as initially perceived.

---

> ### Author Response · Authors · 2024-08-13
>
> We thank the reviewer for taking the time to consider our responses. This is highly appreciated. We also would like to share some more thoughts addressing the last three points.
>
> >In the response, the 1% improvement appears to lack significance, with minimal variation or none at all. This indicates that the novelty compared to previous works is not substantial. Additionally, a 1% enhancement on the Pascal VOC dataset does not represent a significant advancement, especially considering that it is not a particularly challenging dataset.
>
> Regarding novelty... This is harder to argue about since novelty evaluation is naturally subjective. We can only say that in our own opinion (also subjective, of course) the novelty is the property of ideas and concepts, not the results. On the other hand, the significance of our results is that they demonstrate that **complex systems can be outperformed by simple general ideas** that are easy to understand. The "deep" community desperately needs more understanding and our paper contributes in this direction. The value in general unsupervised segmentation ideas studies in our paper is that they can be easily used to design (complex) systems going after SOTA in any practical application. We view our experimental results as a "proof-of-concept" motivating such ideas in general.
>
> We also believe that Pascal is sufficient for a proof-of-concept in WSSS and most related prior work is focused on it. Its "simplicity" and wide use make it harder to achieve any improvement, particularly because (unlike many SOTA methods) we do not use any tricks. Improving over full supervision (even by 1%) also speaks strongly of the power of our simple general ideas. Moreover, we have not seen many examples where the relative performance on PASCAL is not repeated on more complex datasets. Our results on COCO also confirm that simple ideas typically generalize well (better than over-designed systems).
>
> Of course, this is just our subjective opinion.
>
> >The limitation of Potts not functioning effectively on points is a notable drawback.
>
> We agree that point supervision is a drawback for the standard Potts model (and its tight non-convex relaxations, e.g. bilinear). This was a motivation to study convex relaxations, e.g. quadratic. While such relaxations depart from the geometric properties of the boundary motivating the Potts model, consequently they stop suffering from the corresponding boundary shrinking bias that ruins the point supervision. Such relaxations also have their own "probabilistic" motivation (as detailed in "random walker"). These relaxations also have biases, but they are better suited for point supervision.
>
> These were the very motivations for our systematic study of Potts relaxations. We also tried to identify a sweet spot. Our "normalized quadratic" variant came up as such. It is well-motivated conceptually and works best in practice. It is a recommendation in our conclusions.
>
> > scribbles may not be as "weak" as initially perceived.
>
> We agree. However, this conclusion does not diminish the significance of all the hard work made by WSSS subcommunity over the last 10 years allowing one to make this conclusion now. It is amazing to know that labeling only 3% of the pixels works as well as labeling 100% of the pixels and that this quality can be achieved using only simple unsupervised ideas like the Potts model. The main practical significance of our study is that we show that Potts model is sufficient (at least for full scribbles) as prior WSSS work (even if comparable in quality) had to use more complex combinations of losses, system modifications, or post processing steps.

---

### Official Review · Reviewer_QTqT · 2024-07-04

**Soundness:** 3
**Presentation:** 3
**Contribution:** 2
**Rating:** 6
**Confidence:** 3

**Summary:**

This paper considers semantic segmentation under scribble supervision. The paper studies relaxations of the Potts model and proposes a framework for generating soft pseudo-labels, which benefit over hard pseudo-labels in that they can represent uncertainty. The paper highlights problem cases with two standard relaxations, the quadratic and bilinear, and proposes a normalized quadratic relaxation. Moreover, the paper proposes to use a collision cross-entropy loss between the prediction and the pseudo-labels. Different settings are evaluated experimentally, and the proposed approach is compared to the state-of-the-art.

**Strengths:**

Overall, the paper is well written, and the proposed approach is intuitive and should be fairly easy to reproduce, even without code.

The problem under consideration is important as it aims to reduce the manual annotation challenge in image segmentation, which is otherwise costly and time consuming.

The choice of Potts relaxations and cross-entropy terms are supported by experiments, and the proposed approach is further compared to previous methods, showing strong performance.

**Weaknesses:**

Some details are missing or unclear in the main paper. Considering that the soft self-labeling loss in (6) is a key contribution, it would be useful to include some details regarding the optimization of the pseudo-labels in in the main paper from Appendix B. Additionally, pairwise affinities based on intensity edges are mention at line 42, but it is unclear whether they are used in the proposed approach, see questions.

The proposed approach is only evaluated on a single dataset in the main paper. The results on additional datasets in the appendix should be moved to the main paper to better communicate the empirical findings, as they are easy to miss otherwise. This also raises some confusion about Appendix C which says that three datasets are used in Section 3.5, but results are only reported on PASCAL. What was the reason to exclude these from the main paper?

No error bars. Considering that some settings have fairly similar performance, e.g. in Table 3, standard deviation or similar over multiple runs would be useful.

Some figures have excessively small fonts, e.g. Figures 3, 4, 6.

**Questions:**

Are pairwise affinities based on intensity edges used in the proposed approach? If yes, how is w on line 42 defined? If no, how can the model learn to draw the segmentation contours at object borders, which most often correspond to color edges?

How is the bandwidth of the dense neighborhood defined?

Considering that a larger dense neighborhood size reduces pseudo-label quality in Figure 5, did you try values lower than 25? E.g. 1, 2, 5, 10?

**Limitations:**

The paper does not discuss limitations.

Societal impacts are not discussed.

---

> ### Author Rebuttal · Authors · 2024-08-05
>
> **Note:** We can provide only brief answers to the first two questions. We fully understand that NeurIPS reviewers may not be familiar with CRF methodology/terminology. However, please note that it has been standard for image segmentation at least since 1980's with numerous textbooks on the subject (e.g. "Visual Reconstruction" by Blake&Zisserman, "MRF Modeling..."  by Li,  and many more), which may explain why we use such terminology in a relatively relaxed way.
>
> **Are pairwise affinities based on intensity edges used in the proposed approach? If yes, how is w on line 42 defined?**\
> Yes, w defined on 42 is based on the intensity difference of neighboring pixels and we explicitly say this on line 42. The definition is standard and we gave three references [6, 17, 22].
>
> **How is the bandwidth of the dense neighborhood defined?**\
> As the $\sigma$ in the Gaussian kernel over the spatial distance between two pixels on the pixel grid.
>
> **The results on additional datasets in the appendix should be moved to the main paper**\
> The Pascal dataset with scribble supervision is a standard benchmark while the other two datasets are only used with scribble block supervision in [25]. Due to space limitations, we moved these results for the less common datasets to the appendix.
>
> **In Figure 5, did you try neighborhood size lower than 25? E.g. 1, 2, 5, 10?**\
> Yes, we tried the neighborhood sizes 5. In general, the results look similar to the one with NN when the neighborhood size gets smaller.

---

> ### Comment · Reviewer_QTqT · 2024-08-13
>
> Thank you for the answers.
>
> Regarding CRF, I am familiar with the methodology but I do not know all the related work. The source of my confusion was that the formulation between lines 41-42 suggests that there are multiple options in the literature. The equation above line 42 was presented as one possible variant. Thus, my question whether it was used in the proposed approach, which it is now clear that it was. Also, if it is central to the work, the precise definition of w could still be reiterated for completeness, and to capture a broader audience, even if it is standard. Nevertheless, these details are minor.
>
> I have read the other reviews and rebuttals. Overall I retain my original score.

---

> > ### Author Response · Authors · 2024-08-13
> >
> > We now see that we can improve the clarity about the Potts model definition around line 42 and to emphasize that we use this standard "weighted" Potts model. We also totally agree that we should define the function we use for kernel w. While we use standard w based on image contrast [6,17,22] that improves edge alignment in an unsupervised manner, we have to provide this formula for completeness. That would certainly help the readers.
> >
> > Thanks for your feedback!

---

### Official Review · Reviewer_jiv8 · 2024-07-08

**Soundness:** 2
**Presentation:** 2
**Contribution:** 2
**Rating:** 5
**Confidence:** 3

**Summary:**

The work proposes a soft self-labeling framework for weakly supervised semantic segmentation using scribbles. This model-agnostic framework requires only the joint optimization of network predictions and pseudo labels, guided by specific loss functions: collision cross-entropy and log-quadratic Potts relaxations. The design choices are supported by theoretical concepts and experimental results.

**Strengths:**

The work systematically analyzes common loss functions for weakly supervised semantic segmentation. By investigating theoretical concepts and experimental results, it discusses the advantages and disadvantages of these loss functions. Based on this comprehensive analysis, the work concludes by recommending the use of collision cross-entropy and log-quadratic Potts relaxations for a soft self-labeling framework.

**Weaknesses:**

Method
- The proposed method is tested on DeepLab. Although the theoretical concept should hold for any model, its effectiveness on other segmentation models remains unknown.
- The work explains the rationale behind design choices from a theoretical perspective, but it does not clarify why these choices lead to specific pseudo-labels from a vision perspective. For instance, in Figure 5, NN successfully segments the bicycle while DN does not. Is this because the bicycle is a minor class?
- The work does not provide mIoU per category, leaving it unclear whether the method is effective for all categories or just a few major ones.

Minor Writing Issue:
- The legends of the figures, such as Figures 1, 4, and 6, are small.
- Figure 1 (b) contains an extra icon (house).

Related works:
- There are more image-level WSSS works than just [4, 21]. The author should consider including additional relevant works in lines 16-18.

- Weakly-Supervised Image Semantic Segmentation Using Graph Convolutional Networks
- Weakly supervised learning of instance segmentation with interpixel relations.
- Extracting class activation maps from on-discriminative features as well.
- Boundary-enhanced co-training for weakly supervised semantic segmentation.
- Learning pixel-level semantic affinity with image-level supervision for weakly supervised semantic segmentation
- Self-supervised equivariant attention mechanism for weakly supervised semantic segmentation
- Group-wise semantic mining for weakly supervised semantic segmentation

**Questions:**

- Does the same conclusion hold for other segmentation models from an experimental perspective?
- What is the relationship between the visualized pseudo labels/predictions and the specific design choice? In Figure 5, NN successfully segments the bicycle while DN does not. Is this because the bicycle is a minor class? In Figure 7, bilinear tends to over-segment the object instead of under-segmenting it. What is the reason?
- Lines 225-227 mention that the proposed framework can outperform a fully-supervised method with a batch size of 12. Is it also better from a visualization perspective?
- It appears that the performance gap between the proposed framework is much larger on the Cityscapes and ADE20K datasets. Why?

**Limitations:**

I agree with the author that the training time is one of the limitations.
I hope the author can consider my suggestion in the above section to build a link between the design choice and the pseudo labels from the computer vision perspective.

---

> ### Author Rebuttal · Authors · 2024-08-05
>
> **Does the same conclusion hold for other segmentation models from an experimental perspective?**\
> We do not see any technical reasons to expect significant differences compared to current results, which are consistent with their theoretical properties shown in our paper.
>
> **Results for other segmentation models**\
> We showed comparison results on different models in Figure 6 (MobileNetV2) and in Table 5 (ResNet101).
>
> **What is the relationship between the visualized pseudo labels/predictions (in Figure 5) and the specific design choice of Potts model relaxation? In Figure 7, bilinear tends to over-segment the object instead of under-segmenting it. What is the reason?**\
> As for the neighborhood system choice, lines 202-204 explain the failure of DN shown in Figure 5 and refer to paper [37] for detailed technical reasoning. Figure 5 uses the same quadratic relaxation for NN and DN. In Figure 5, NN shows better edge alignment compared to DN, as conceptually explained in [37].
>
> As for Potts relaxations, compared in Figure 7, our detailed discussion in Section 2.1 and Figure 1 illustrate the conceptual properties of different Potts relaxations. Specific to your question, the bilinear relaxation tends to produce over-confident (hard) labeling and easily gets stuck in the local minima. For example, bilinear pseudo-labeling results look like the hard version of the initialization in Figure 7 (a). This is not exactly over- vs under-segmentation, in our opinion.
>
> **the performance gap between the proposed framework is much larger on the Cityscapes and ADE20K datasets**\
> We assume you mean the gap between the proposed framework and the fully-supervised method. It is because these two datasets are more difficult compared to Pascal, in part due to the larger number of classes. This gap is even larger for other WSSS methods. If you mean some other gap, please clarify.

---

> > ### Comment · Reviewer_jiv8 · 2024-08-08
> >
> > Thank the authors for the rebuttal and the comments from other reviewers.
> >
> > I have read the other reviews and feel that some of my concerns are not fully addressed.
> >
> > I am not satisfied with the answer to the concern, "Does the same conclusion hold for other segmentation models from an experimental perspective?" After reading the answer, the impact and improvement on other models is still unknown. Additionally, it looks like other reviewers (dJES) have the same question (q.9).
> >
> > I believe the concern, "The work does not provide mIoU per category, leaving it unclear whether the method is effective for all categories or just a few major ones," is important. Unfortunately, the authors do not respond to this.
> >
> > The concern, "Lines 225-227 mention that the proposed framework can outperform a fully-supervised method with a batch size of 12. Is it also better from a visualization perspective?" is also not addressed. dJES has a similar concern in q.7.
> >
> > The authors don't promise to address the minor writing issues, which QTqT also mentions. It is okay to see this in a first draft, but for the camera-ready version, these minor writing issues are terrible.
> >
> > I decide to lower my score at this moment, but I may adjust accordingly after seeing more discussion.

---

> ### Author Response · Authors · 2024-08-12
> **Authors' response to the additional comments by Reviewer jiv8**
>
> > I have read the other reviews and feel that some of my concerns are not fully addressed.
>
> We are sorry to hear this and will try to do better in this second iteration. As you noted, some of dJES concerns overlap with yours, but we did not realize you might not see our response to dJES. We will rectify this below, though our screen now looks like the same discussion is copied around. Anyways...
>
> > I am not satisfied with the answer to the concern, "Does the same conclusion hold for other segmentation models from an experimental perspective?" After reading the answer, the impact and improvement on other models is still unknown. Additionally, it looks like other reviewers (dJES) have the same question (q.9).
>
> We did not have any concerns about ViT in our paper as Potts is a general unsupervised loss for segmentation and its properties for training are independent of architecture. We see no technical reason to assume that Potts would not matter for ViT. While it might be a better architecture, it is not smart by itself without training.
>
> However, your original review and dJES commens motivated us to add ViT experiments. By now we obtained the following results for ViT (vit_base_patch16_224) on PASCAL with full scribbles. We hope it helps to confirm that Potts matters for WSSS with any backbones, including ViT. The results below show that Potts matters for our approach using it during training ViT, and it matters for [3] (arXiv from Marcvh 2024, reference from dJES's review) using dense CRF (a version of Potts) to post-process ViT output.
>
> **batch size 12**:
>
> only partial CE  =>  **74.61%** mIOU (**the baseline [3] suggested by  dJES**)
>
> partial CE + Log Div Potts +  CCE => **80.80%** mIOU   (**our**)
>
> **batch size 16**:
>
> only partial CE   =>  **75.10%** mIoU  (**the baseline [3] suggested by  dJES**)
>
> partial CE + Log Div Potts + CCE => **80.94%** mIOU   (**our**)
>
> **arXiv [3], March 2024 (no batch size reported)**:
>
> =>  **78.7%** mIoU  (**without CRF postprocessing**)
>
> =>   **80.3%** mIoU  (**with CRF post-processing**)
>
> We will be happy to add these numbers to the final version of Table 5 and discuss the importance of the Potts model across architectures, including ViT. Our final numbers for ViT (with batches 12 and 16) might be higher than the above as we might find better tuning (learning rate, number of epochs, etc) when we have a bit more time.
>
> Another related general observation about the numbers above. It is natural to expect that direct integration of the Potts model as a loss for standard end-to-end training may be a simpler and more stable approach than using the Potts model for postprocessing, as in [3]. The latter requires independent tuning of multiple stages.
>
> >I believe the concern, "The work does not provide mIoU per category, leaving it unclear whether the method is effective for all categories or just a few major ones," is important. Unfortunately, the authors do not respond to this.
>
> While this is an interesting suggestion, class-specific mIOU is not too common in related WSSS literature. It did not occur to us to collect such statistics primarily because we will have very little to compare against as most of the relevant prior WSSS works (cited in our submission) report only the average over all classes. We hope you will find this excuse acceptable.
>
> >The concern, "Lines 225-227 mention that the proposed framework can outperform a fully-supervised method with a batch size of 12. Is it also better from a visualization perspective?" is also not addressed. dJES has a similar concern in q.7.
>
> About "visualization perspective". If you mean qualitative results, of course, it is easy to "cherry-pick", as common. It is possible even with 1% improvement over full supervision. However, we did not think it was important (given the 1% difference) and preferred to save this space for numerical results. We can reconsider this.
>
> There is a natural explanation for why WSSS could beat full supervision. We observed that there are sufficiently many wrong labels in the original Pascal ground truth masks. WSSS may avoid overfitting to such errors. This point may be interesting enough to squeeze into the paper, but it is highly speculative, and (similarly to the qualitative improvements above) we are not sure it is worth the space it would require to illustrate all of these. Let us know what you think.
>
> > The authors don't promise to address the minor writing issues, which QTqT also mentions. It is okay to see this in a first draft, but for the camera-ready version, these minor writing issues are terrible.
>
> Sorry. We assumed that the focus of the discussion phase was to answer questions and concerns posted by reviewers. Of course, we always use all the corrections/typos and other minor writing issues found by the reviewers to correct the final version. Sorry if the lack of acknowledgment made it look like we plan to ignore them. Thanks a lot for finding all of these!

---

> ### Comment · Reviewer_jiv8 · 2024-08-13
>
> I thank the authors for providing new results. Please give me some time to reconsider the per-category mIoU part.

---

> ### Comment · Reviewer_jiv8 · 2024-08-13
>
> I appreciate the patience of the authors.
>
> I understand that the authors may feel it is unnecessary to include per-category mIoU. They may also feel it is unfair because many other works do not include it. While this reasoning is acceptable to me, it is just fine. I have decided to give a borderline accept.
>
> Regarding the issue of the ground truth in the Pascal dataset, I think the explanation is helpful for readers. However, since there is no evidence provided in the paper to support this claim, I suggest using a less strong tone when mentioning this explanation.

---

> > ### Author Response · Authors · 2024-08-13
> >
> > We thank the reviewer for taking the time to carefully consider our response.
> >
> > We also agree that the tone about the ground truth on Pascal should be carefully weighted. We will revisit this part.

---

### Official Review · Reviewer_Pwad · 2024-07-08

**Soundness:** 3
**Presentation:** 4
**Contribution:** 3
**Rating:** 5
**Confidence:** 3

**Summary:**

This paper proposes a new framework for Weakly-Supervised image Segmentation. The main contribution is to use a soft-labelling approach which is considered superior to classic hard labelling because it can keep track of the centanty of the label. Then different forms of second order potts relaxation and cross-entropy are evaluated.
Results show that the proposed approach with its best setting is able to perform better than previous approaches and comparably to a fully supervised approach with only 3% of annotated pixels.

**Strengths:**

\+ The paper is in general well written and easy to follow

\+ Nice illustrations help to understand the proposed approaches

\+ Ablations on the most important components help to understand the significance of each component

**Weaknesses:**

\- It is not clear if results are significant. For instance is 1% a meaningful improvement or it can be generated by just noise? It is important to see results on multiple runs with also standard deviation.

\- The motivation about soft-labelling because it keeps information about the label certainty is not clear to me

\- In tab. 5 it is not clear what is the base model you start from. Could you report also the base model with standard cross-entropy and hard potts model? It seems that results are very good because the baseline is already quite good.

\- Authors did not say much about the computational cost of the proposed approach compared to common models.

**Questions:**

I wrote some questions in the weaknesses part.

**Limitations:**

Authors did not mention the possible limitations of the proposed approach.
One possible limitation is the need of additional hyper-parameters to tune which can take time. Authors should comment on that.

---

> ### Author Rebuttal · Authors · 2024-08-05
>
> **The motivation for using soft pseudo-labels is not clear**\
> First, soft pseudo-label (distribution) is a generalization of the standard hard pseudo-label (one-hot distribution). Softness can represent more information, which can be naturally interpreted probabilistically as the uncertainty of the random variable representing an unknown true class label. For example, soft pseudo-label in the form of a uniform distribution implies no knowledge of the true class, while one-hot implies that the true class is known with certainty. Standard hard pseudo-labels can only represent the latter.
>
> Second, soft pseudo-labels naturally appear in the context of the Potts relaxations that significantly expand the class of unsupervised losses that could be used for weakly supervised segmentation. Such relaxations are the main focus of this paper.
>
> Third, soft pseudo-labels also motivate the study of different forms of the cross-entropy term in self-labeling losses.
> Most related prior work use standard cross-entropy with hard pseudo-labels replacing hard ground truth labels, as in fully supervised methods. Our focus on soft pseudo-labels opens a question whether standard-cross entropy is the right choice for uncertain targets. This question also has a very specific numerical motivation discussed in Figure 3.
>
> **Missing baseline with standard cross-entropy and hard Potts model**\
> We showed such result in Table 5 and the method is "GridCRF loss [27]".
>
> **Computational cost of the proposed approach compared to common models**\
> We provide the computational cost compared to the method [36] on line 428. In general, our computational costs are on par with or better than related self-labeling methods for WSSS, e.g. [26, 27].
>
> **Is 1% a meaningful improvement or just noise? It is important to see results on multiple runs with also standard deviation.**
>
> While we did not properly collect the variations over multiple runs (the tests are expensive even with out this), our informal observations are that this variation is very low (below 1%). Also, it is standard in WSSS literature (and prior works we cite) to repost the best run.
>
> Some unsolicited discussion of the results:
>
> For completeness, Table 5 includes many prior scribble segmentation methods, including those that design specialized complex architectures, see the "architectural modifications" block. It makes the most sense to directly compare our results only with methods using standard architectural backbones (the last block in Table 5) since this constitutes a fair comparison of different loss functions for WSSS, which is the focus of our study. For example, one can easily use such general losses, including ours, to build complex systems or specialized architectures, but this is not the focus of our work studying the basic conceptual properties of a large general class Potts relaxations.
>
> Indeed, according to Table 5, our method with standard V3+ backbone (16 batches) outperforms the method in [25] modifying V3+ architecture (also 16 batches) only by 1%, which may or may not be significant. However, it may not be a fair comparison since [25] designs a more complex 2-branch architecture. Moreover, their approach has some technical flaws as their training is not guaranteed to converge (their procedurally-defined iterative method does not have a clearly defined self-labeling loss). Such ad-hoc methods typically do not generalize well to datasets other than those for which they were designed (e.g. Pascal in this case).
>
> In any case, it makes more sense to compare our results mainly within the last block where we collected many 12-batch results on V3+ from comparable prior work studying loss functions on standard architectures. We consistently outperform those by at least 2% or more only by using new loss functions and a stronger well-defined optimization algorithm. In this 12-batch scenario we even outperform the full supervision by 1%. One of such experiments may or may not be significant, but the consistency of our improvements matters particularly because they come only from simple general loss functions that anyone can use in any system or architecture (simple or complex).

---

### Decision · Program_Chairs · 2024-09-25

**Decision:**

Reject

**Comment:**

This submission proposes a method for weakly-supervised segmentation using scribbles. The submission proposes to use a soft pseudo-labelling approach that jointly optimizes pseudo-labels and predictions.
On the positive side, several reviewers mention that:
1. The paper is well written and easy to understand
2. It proposes a well-motivated pseudo-labelling method, with limitations that are reasonably well-discussed.

At the same time, after the discussion period, all reviewers continued to have concerns about the marginal results and the lack of statistical significance testing.

In addition, there were concerns about the quantitative evaluation and the authors’ stance that “we leave the comprehensive evaluation on more different tasks and more datasets to the future work”. Reviewers would have liked to see a more comprehensive analysis, and to see some of the experiments in the Appendix moved to the main paper, but the authors were reluctant to do this.

Ultimately, the submission received mixed recommendations from the reviewers. The AC believes that the concern about the statistical significance of the approach is an important one. Given that, as the authors mention, their “work is focused on studying the properties of different relaxations of the standard Potts model [...], as well as different cross-entropy terms”, the experimental validation is key. The AC recommends rejection at this time, and encourages the authors to implement the suggestions of the reviewers to extend the evaluation of this work in its next iteration.